# Contractile acto-myosin network on nuclear envelope remnants positions human chromosomes for mitosis

Alexander JR Booth[1], Zuojun Yue[1], John K Eykelenboom[1], Tom Stiff[2], GW Gant Luxton[3], Helfrid Hochegger[2], Tomoyuki U Tanaka[1]*

[1]Centre for Gene Regulation and Expression, School of Life Sciences, University of Dundee, Dundee, United Kingdom; [2]Genome Damage and Stability Centre, University of Sussex, Brighton, United Kingdom; [3]College of Biological Sciences, University of Minnesota, Minneapolis, United States

**Abstract** To ensure proper segregation during mitosis, chromosomes must be efficiently captured by spindle microtubules and subsequently aligned on the mitotic spindle. The efficacy of chromosome interaction with the spindle can be influenced by how widely chromosomes are scattered in space. Here, we quantify chromosome-scattering volume (CSV) and find that it is reduced soon after nuclear envelope breakdown (NEBD) in human cells. The CSV reduction occurs primarily independently of microtubules and is therefore not an outcome of interactions between chromosomes and the spindle. We find that, prior to NEBD, an acto-myosin network is assembled in a LINC complex-dependent manner on the cytoplasmic surface of the nuclear envelope. This acto-myosin network remains on nuclear envelope remnants soon after NEBD, and its myosin-II-mediated contraction reduces CSV and facilitates timely chromosome congression and correct segregation. Thus, we find a novel mechanism that positions chromosomes in early mitosis to ensure efficient and correct chromosome–spindle interactions.
DOI: https://doi.org/10.7554/eLife.46902.001

*For correspondence:
t.tanaka@dundee.ac.uk

**Competing interests:** The authors declare that no competing interests exist.

## Introduction

The mitotic spindle is composed of microtubules (MTs) and plays central roles in chromosome segregation during mitosis. To ensure proper chromosome segregation, chromosomes must be efficiently captured and aligned on the mitotic spindle prior to their segregation (*Godek et al., 2015*; *Tanaka, 2010*; *Maiato et al., 2017*). For this, chromosomes establish initial interaction with spindle MTs soon after nuclear envelope breakdown (NEBD). Subsequently, sister chromatids interact with MTs from opposite spindle poles (chromosome bi-orientation).

Several recent studies have revealed how chromosome–MT interactions are established in early mitosis and how these interactions determine the positions of chromosomes on the mitotic spindle. For example, chromosomes initially interact with the lateral side of spindle MTs, and the consequent positioning of the chromosomes facilitates their bi-orientation (*Magidson et al., 2011*; *Itoh et al., 2018*). Bi-orientation itself causes chromosomes to move to the middle of the mitotic spindle. These movements could also occur prior to bi-orientation, facilitated by chromosome sliding along preexisting spindle MTs (*Kapoor et al., 2006*; *Barisic et al., 2014*).

However, little is known about how chromosome positions are regulated independently of spindle MTs in early mitosis and what the biological significance of such regulation might be. For example, the positions of chromosomes relative to the spindle poles may affect the efficiency of establishing chromosome interaction with spindle MTs soon after NEBD. If chromosomes were located far away from the spindle poles, more time would be required to establish the initial

chromosome–MT interaction. Moreover, should chromosomes be located behind a spindle pole, that is opposite from the bulk of spindle and outside of the pole-to-pole region, their interaction with MTs from the other spindle pole would be significantly delayed.

## Results

### Chromosome scattering volume is reduced in early prometaphase, independently of spindle MTs

To investigate chromosome positioning during early mitosis, we quantified how widely chromosomes were scattered in space. To do this, we acquired three-dimensional images of fluorescently labeled DNA (SiR-DNA) in live mitotic human U2OS cells. Subsequently, we obtained the convex hull of the chromosome distribution. The convex hull in two and three dimension (2D and 3D) is the minimal polygon and polyhedron, respectively, that wrap multiple objects (chromosomes in this case). This is schematically shown in *Figure 1A*, using a shrinking 'rubber band' and 'balloon' for intuitive demonstration. We determined the volume of this minimal polyhedron or shrunk balloon that wrapped chromosomes, and defined it as the chromosome scattering volume (CSV), which represents how widely chromosomes scatter in 3D space.

To synchronize U2OS cells in early mitosis, we used a previously described U2OS cell line in which the *CDK1* gene was replaced, by genome engineering, with *cdk1-as,* whose gene product Cdk1-as can be specifically inhibited by an ATP analogue 1NM-PP1 (*Rata et al., 2018*). The U2OS *cdk1-as* cells were arrested in G2 with 1NM-PP1, and subsequently released by 1NM-PP1 washout to synchronously undergo mitosis. NEBD was identified by the release of the nuclear-localizing fluorescent reporter (GFP-LacI-NLS) into the cytoplasm (*Figure 1—figure supplement 1*). Using these methods, we quantified the CSV in synchronized U2OS *cdk1-as* cells as they progressed from NEBD (defined as t = 0) into prometaphase. We found that CSV was prominently reduced within the first 8 min following NEBD, and reduction continued more slowly over the following ~10 min (*Figure 1B*). CSV reduction was also observed in asynchronous, wild-type U2OS cells (*Figure 1—figure supplement 2*), indicating that it was not an artifact caused by *cdk1-as*-mediated cell cycle synchronization. Consequently, we used U2OS *cdk1-as* cells for synchronous entry into mitosis for the rest of the experiments presented in this work, unless otherwise stated.

It is possible that the CSV reduction observed following NEBD was an outcome of the interaction between chromosomes and spindle MTs. To test this possibility, we quantified CSV in cells treated with nocodazole. After nocodazole treatment, MTs were almost completely depleted (*Figure 1—figure supplement 3*). In control cells, chromosomes moved inward after NEBD, and subsequently aligned on the metaphase plate (*Figure 1C*, top-row images; *Video 1*). In cells lacking visible MTs, chromosomes also moved inward after NEBD, but then remained in a spherical formation (*Figure 1C*, bottom-row images; *Video 1*; *Figure 1—figure supplement 4*). The CSV was reduced with very similar kinetics in both the presence and absence of MTs (*Figure 1C*, graphs). CSV may be slightly smaller at 4 min following NEBD in control cells; if so, this could be due to mild compression of the nuclear envelope (NE) remnants caused by the rapid MT-dependent inward movement of centrosomes (*Figure 1C*, image at +4 min in control). Consequently, we conclude that the overall CSV reduction observed following NEBD is not an outcome of chromosome interaction with spindle MTs.

Alternatively, chromosome condensation, which occurs in early mitosis, may cause the CSV reduction after NEBD. To test this possibility, we quantified the total volume of chromosome mass (chromosome mass volume [CMV]; *Figure 1—figure supplement 5*) after NEBD. Unlike the CSV, the CMV did not considerably change following NEBD. Thus, we conclude that the reduction in CSV was not due to chromosome condensation, which likely occurs mainly before NEBD (*Mora-Bermúdez et al., 2007*).

### Actin accumulates outside of the NE in prophase in a LINC complex-dependent manner, and the actin network shrinks after NEBD

What mechanisms, then, might promote CSV reduction after NEBD? The CSV reduction could be explained if chromosomes were surrounded by a physical barrier that contracts following NEBD. Consistent with this model, we found that in early mitosis an actin network accumulated on the NE, peaking in intensity around NEBD and persisting after it (*Figure 2A*; *Video 2*). The volume inside

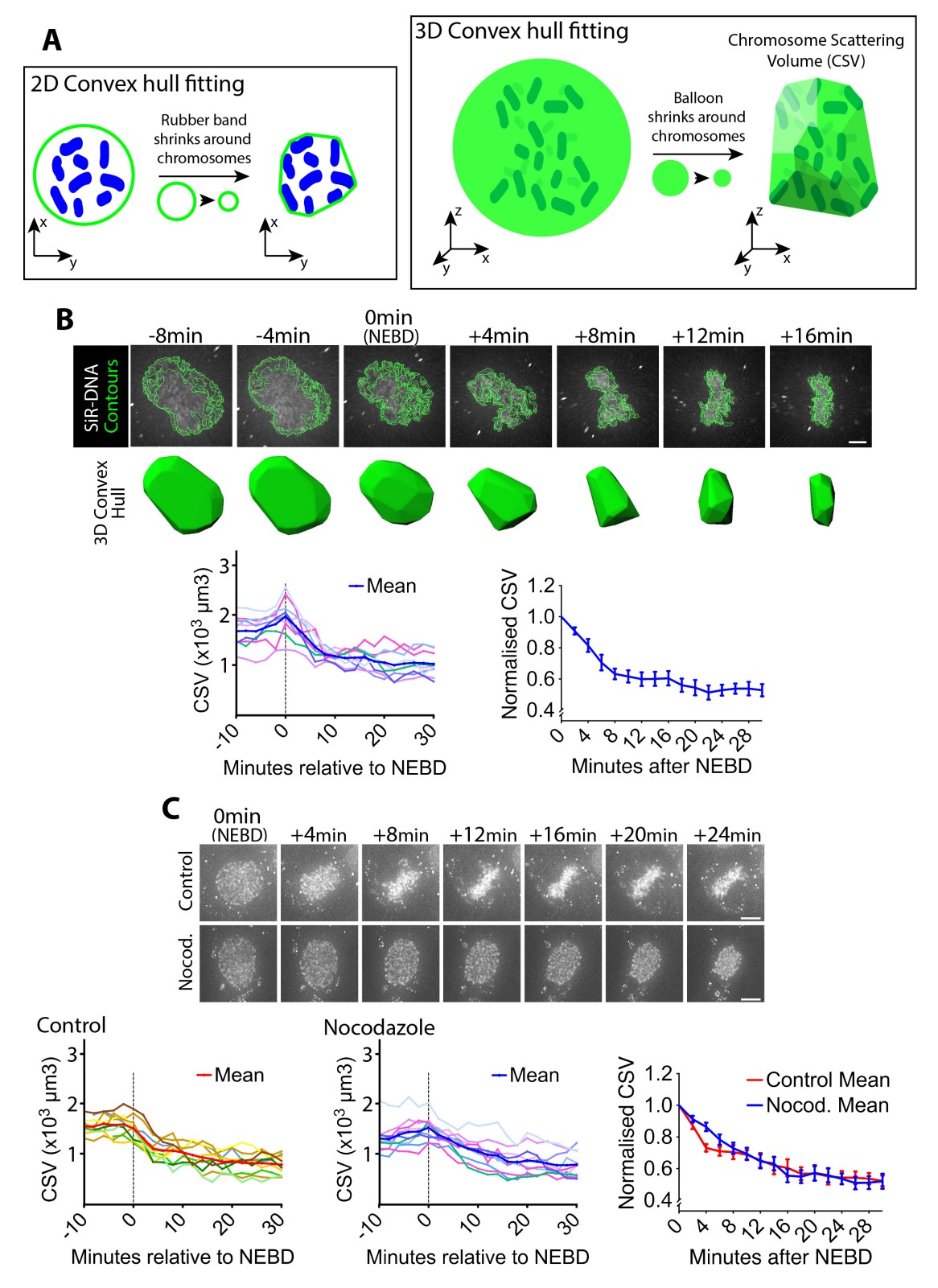

**Figure 1.** Chromosome scattering volume is reduced in early prometaphase, independently of spindle MTs. (**A**) Diagrams show how the chromosome scattering volume (CSV) was defined. Left-hand diagram shows how the convex hull was generated in two dimensions (2D) to represent chromosome distribution. A 'rubber band' (green) shrinks around chromosomes (blue) in 2D to create a convex hull or a minimum polygon wrapping chromosomes. Right-hand diagram shows how the convex-hull was generated in three dimensions (3D) to represent chromosome distribution. A 'balloon' (green)

*Figure 1 continued on next page*

*Figure 1 continued*

shrinks around chromosomes (dark green) in 3D to create a convex hull or a minimal polyhedron wrapping chromosomes. The volume of the 3D convex hull (chromosome scattering volume; CSV) quantifies how widely chromosomes are scattered in space. (B) CSV decreases immediately after NEBD. Images (top) are z-projections (z-sections are projected to 2D images) of a representative cell stained with SiR-DNA to visualize chromosomes (white) alongside their perimeter contours at z-sections (green lines; see Materials and methods). Time is shown, relative to NEBD. Timing of NEBD was determined by observation that GFP-LacI-NLS spread out of the nucleus (*Figure 1—figure supplement 1*). Scale bars, 6 μm. Bottom shows corresponding CSV (green). The left-hand graph shows CSV measurements in individual cells aligned by the time relative to NEBD (n = 9). To make the right-hand graph, these CSV values were normalized to the CSV value at NEBD in each cell, and the mean of the normalized CSV values were plotted at each time point. Error bars, s.e.m. (C) CSV decreases after NEBD even in the absence of MTs. Images (z-projections) show representative cell with SiR-DNA-stained chromosomes, which entered mitosis in the presence of 3.3 μM nocodazole (Nocod., bottom) or DMSO (control, top). 0.5 μM MK-1775 Wee1 inhibitor was used in both conditions to allow cells to enter mitosis. Time is shown relative to NEBD. Timing of NEBD was determined by observation that GFP-LacI-NLS spread out of the nucleus (*Figure 1—figure supplement 1*). Scale bars, 10 μm. Left-hand and center graphs show CSV measurements in individual control and nocodazole-treated cells, respectively (n = 10 each). The right-hand graph compares the means of normalized CSV between the nocodazole-treated and control cells (error bars, s.e.m.), as in the right-hand graph in (B). Reduction of normalized CSV was not significantly different between control and nocodazole treatment, when all the time points were considered by two-way ANOVA (p=0.81). Nonetheless, normalized CSV was significantly different between the two groups at +4 min (t-test, p=0.0016), but not at other time points.

DOI: https://doi.org/10.7554/eLife.46902.002

The following source data and figure supplements are available for figure 1:

**Source data 1.** Data at individual time points in individual cells.
DOI: https://doi.org/10.7554/eLife.46902.008
**Figure supplement 1.** Method for estimation of the NEBD timing.
DOI: https://doi.org/10.7554/eLife.46902.003
**Figure supplement 2.** CSV decreases and actin accumulates on the NE around NEBD in asynchronous wild-type U2OS cells (without *cdk1-as*).
DOI: https://doi.org/10.7554/eLife.46902.004
**Figure supplement 3.** Confirmation that MTs are absent after nocodazole treatment.
DOI: https://doi.org/10.7554/eLife.46902.005
**Figure supplement 4.** The chromosome distribution in three-dimensional space with and without nocodazole treatment.
DOI: https://doi.org/10.7554/eLife.46902.006
**Figure supplement 5.** Chromosome mass volume did not considerably change after NEBD.
DOI: https://doi.org/10.7554/eLife.46902.007

this actin network was rapidly reduced within 6 min following NEBD (*Figure 2—figure supplement 1*), suggesting the contraction of the network. During this time, the chromosomes were surrounded by, and the outermost chromosomes were found immediately adjacent to, the actin network (*Figure 2—figure supplement 2*; *Video 3*). We used *cdk1-as* in these experiments to synchronize cells, however the actin network was also observed in wild-type U2OS cells (*Figure 1—figure supplement 2*, yellow arrowheads), indicating that it is not an artifact due to *cdk1-as* regulation.

We investigated in more detail the localization of the actin network on the NE or its remnants, following NEBD. Using Airyscan super-resolution microscopy, we compared the localization of the actin network and the lamin B1 network; the latter is found on the nucleoplasmic surface of the NE (*Ungricht and Kutay, 2017*) and remains substantially intact soon after NEBD (*Georgatos et al., 1997*). The actin network was separated outwardly from lamin B1 by 150–200 nm (*Figure 2B*), suggesting that it may localize on the cytoplasmic surface of the NE (or its remnants) following NEBD. A candidate receptor for the actin network on the NE is the LINC (linker of nucleoskeleton and cytoskeleton) complex, one end of which is anchored in the NE while the other end extends to the cytoplasm and interacts with actin (*Luxton and Starr, 2014*). To test whether the NE-association of the actin network during early mitosis was LINC complex-dependent, we visualized the actin network in

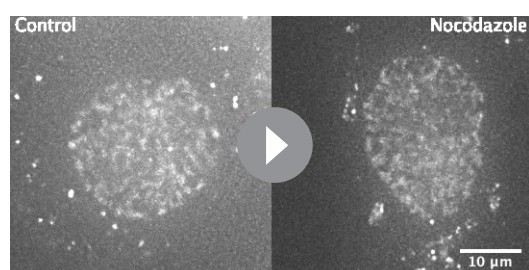

**Video 1.** CSV decreases after NEBD even in the absence of MTs. Video of representative cells (Control and Nocodazole-treated) shown in *Figure 1C* (SiR-DNA staining). Frame intervals are 2 min. Video is shown at 600x speed.
DOI: https://doi.org/10.7554/eLife.46902.009

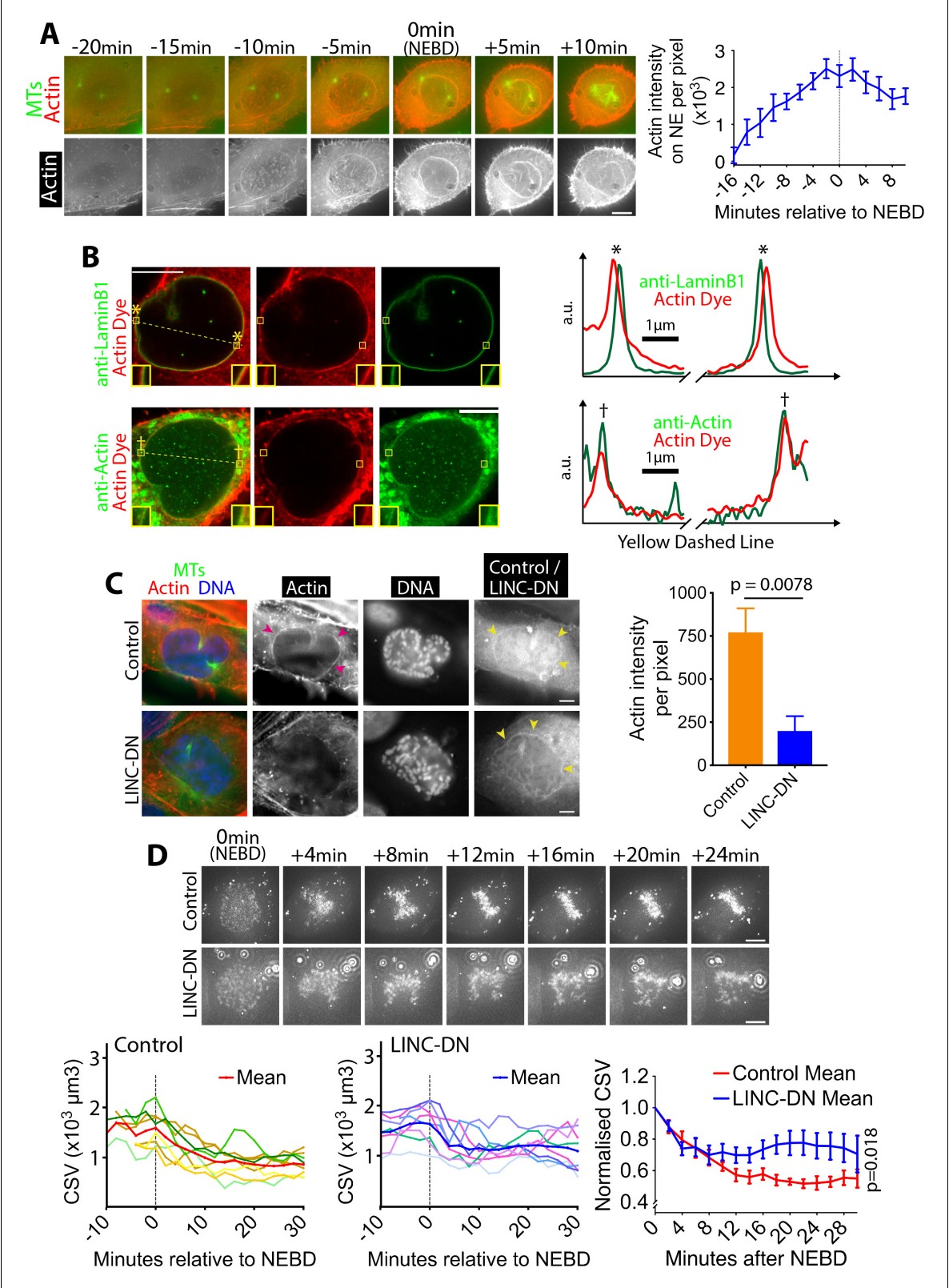

**Figure 2.** Actin accumulates outside of the NE in prophase, and its network shrinks after NEBD. (**A**) Actin accumulates on the NE around NEBD. Images (z-projections) show a representative cell expressing GFP-tubulin and mCherry-Lifeact (that fluorescently marks F-actin). Time is relative to NEBD. Timing of NEBD was determined by the influx of cytoplasmic GFP-tubulin into the nucleus. Scale bars, 10 μm. Graph shows mean Lifeact fluorescence intensity (per pixel) around the nucleus over time (n = 8; error bars, s.e.m). (**B**) Actin localizes outside of the NE. Images show single super-resolution

*Figure 2 continued on next page*

*Figure 2 continued*

z-sections of cells, which were fixed and stained for actin with phalloidin (actin dye; red). Cells were also immunostained (green) with anti-Lamin B1 (top) or anti-actin antibody (bottom), respectively. The same secondary antibody was used for the two immunostainings. These cells were undergoing chromosome compaction (confirmed by DNA staining). Insets show magnification of the regions in yellow boxes. Scale bar, 10 μm. Graphs show intensity of actin dye (red) and immunostaining (green) along the dashed yellow lines in left images (middle parts are omitted). The peaks in graphs, marked with asterisks (top) and daggers (bottom), locate at the regions in yellow boxes in left images. a.u, arbitrary unit. (C) The LINC complex is required for accumulation of the actin network. Images show single z-sections of representative cells, expressing either an RFP-tagged LINC-DN construct (SR-KASH) or an RFP-tagged control construct (KASHΔL) (*Luxton et al., 2010*). Cells were stained for actin with phalloidin (red), DNA with Hoechst 33342 (blue) and MTs with anti-tubulin antibody (green). Right-hand-most images show RFP signals showing localisation of the LINC-DN or control construct. To focus on cells in prophase or early prometaphase, we analyzed cells where chromosome compaction had started and the centrosomes had separated, but bipolar spindle formation had not yet been completed. MK-1775 Wee1 inhibitor was used to make the same condition as in (D). Magenta arrowheads indicate the actin network on NE in control. Yellow arrowheads indicate localization of LINC-DN and its control on NE. Scale bars, 10 μm. Graph on right shows mean intensity of actin signal around the nucleus in cells expressing LINC-DN vs a control construct (control n = 18, LINC-DN n = 10). *p* value was obtained by *t*-test. Error bars, s.e.m. D)Removal of the actin network results in incomplete CSV reduction after NEBD. Images (z-projections) show a cell with SiR-DNA-stained chromosomes, expressing either LINC-DN or a control construct. Localization of LINC-DN and its control was as (C). MK-1775 was used in both conditions to allow cells to enter mitosis. Time is relative to NEBD. Timing of NEBD was determined by observation that GFP-LacI-NLS spread out of the nucleus (*Figure 1—figure supplement 1*). Scale bars, 10 μm, Left-hand and center graphs show CSV measurements in individual cells expressing control and LINC-DN constructs, respectively (n = 8 each). The right-hand graph compares the means of normalized CSV (as in *Figure 1B*). Error bars, s.e.m. *p* value (control vs LINC-DN) was obtained by two-way ANOVA.
DOI: https://doi.org/10.7554/eLife.46902.010

The following source data and figure supplements are available for figure 2:

**Source data 1.** Data at individual time points in individual cells.
DOI: https://doi.org/10.7554/eLife.46902.013
**Figure supplement 1.** The volume inside of the actin network is rapidly reduced after NEBD.
DOI: https://doi.org/10.7554/eLife.46902.011
**Figure supplement 2.** During contraction of the actin network, outermost chromosomes are located right beneath the network.
DOI: https://doi.org/10.7554/eLife.46902.012

cells expressing a previously described dominant negative construct of the LINC complex (LINC-DN) or its control – SR-KASH and KASHΔL, respectively, reported in (*Luxton et al., 2010*). As expected, both constructs localized to the NE (*Figure 2C*, yellow arrowheads). Crucially, the fluorescence intensity of the actin network was significantly diminished in cells expressing LINC-DN, compared with cells expressing its control (*Figure 2C*, magenta arrowheads, graph). Therefore, the actin network localizes to the cytoplasmic surface of the NE and its remnants, following NEBD, in a LINC complex-dependent manner.

If the actin network were a physical barrier, confining chromosomes, and its contraction were to lead to the CSV reduction after NEBD, LINC complex disruption should also alleviate CSV reduction. We tested this hypothesis by measuring the CSV in cells expressing LINC-DN and its control (*Figure 2D*; *Video 4*) and found that the CSV reduction was incomplete after NEBD with the LINC-DN expression, compared with the control expression. Thus, it is likely that the actin network on NE remnants confines chromosomes and that its contraction leads to CSV reduction.

## Myosin II activity is required for actin network contraction and CSV reduction after NEBD

We next addressed how the actin network contracts on the NE remnants to reduce the CSV after NEBD. We first tested whether actin depolymerization causes the CSV reduction. An actin

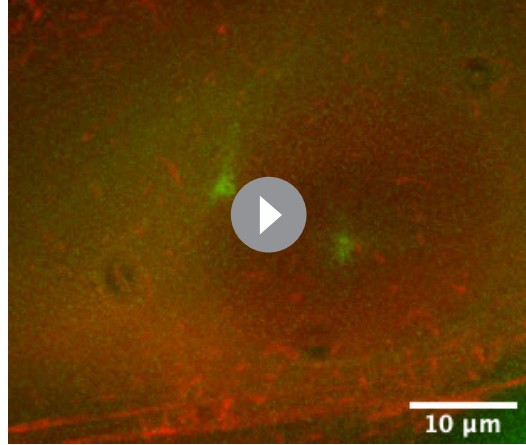

**Video 2.** Actin accumulates on the NE around NEBD. Video of representative cells shown in *Figure 2A* (expressing GFP-tubulin and mCherry-Lifeact). Frame intervals are 1 min. Video is shown at 600x speed.
DOI: https://doi.org/10.7554/eLife.46902.014

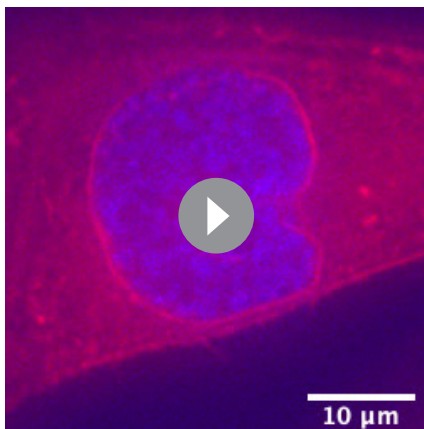

**Video 3.** During contraction of the actin network, outermost chromosomes are located right beneath the network. Video of representative cells shown in *Figure 2—figure supplement 2* (expressing mCherry-Lifeact, GFP-tubulin and H2B-cerulean). GFP-tubulin is not shown in the Video. Frame intervals are 1 min. Video is shown at 600x speed.
DOI: https://doi.org/10.7554/eLife.46902.015

**Video 4.** Removal of the actin network results in slower CSV reduction after NEBD. Video of representative cells (Control and LINC-DN expressing) shown in *Figure 2D* (SiR-DNA staining). Frame intervals are 2 min. Video is shown at 600x speed.
DOI: https://doi.org/10.7554/eLife.46902.016

depolymerization inhibitor jasplakinolide had no effect on CSV reduction after NEBD (*Figure 3—figure supplement 1*), although it successfully inhibited subsequent cytokinesis as previously reported (*Murthy and Wadsworth, 2005*). Thus it is unlikely that actin depolymerization promotes actin network contraction or CSV reduction.

The actin-dependent motor protein myosin II is a major mediator of acto-myosin contractility in non-muscle cells (*Svitkina, 2018*). Thus, we next tested the possible involvement of myosin II in the actin network contraction and CSV reduction. We found that myosin II co-localized with the actin network on the NE (*Figure 3—figure supplement 2*, yellow arrowheads). Given this, we next inhibited myosin II activity using para-nitroblebbistatin (pnBB), a photo-stable version of the myosin II inhibitor blebbistatin, which is suitable for live-cell imaging (*Képiró et al., 2014*). Notably, pnBB treatment alleviated both the CSV reduction (*Figure 3A*; *Video 5*) and the actin network contraction (*Figure 3—figure supplement 3A,B*) following NEBD, which suggests that these processes are dependent on the myosin II activity.

On the other hand, the pnBB treatment did not affect CMV (see *Figure 1—figure supplement 5*) after NEBD (*Figure 3—figure supplement 4*). This suggests that, while myosin II activity facilitates CSV reduction, this reduction is not due to enhanced chromosome condensation. Moreover, pnBB treatment did not significantly change the distance between spindle poles soon after NEBD (*Figure 3—figure supplement 3A,C*). As far as we observed, pnBB-treated cells did not seem to show considerably altered shapes of the spindle (*Figure 3—figure supplements 3* and *5*), compared with control (and LINC-DN-expressing) cells (*Figure 3—figure supplement 6*), following NEBD. Thus, it seems unlikely that the alleviated CSV reduction due to pnBB treatment was the outcome of an abnormal mitotic spindle. Nonetheless, pnBB treatment often caused the actin network to extend beyond the spindle poles at the initiation of spindle formation (*Figure 3—figure supplement 5*, pnBB, 5–20 min).

Furthermore, the pnBB treatment alleviated the CSV reduction in the presence of nocodazole (*Figure 3B*; *Video 6*), indicating that myosin II promotes actin network contraction and CSV reduction after NEBD in a spindle MT-independent manner. Double treatment with pnBB and nocodazole seemed to further alleviate the CSV reduction during 0–6 min, compared with pnBB treatment alone (compare *Figure 3A and B*). Thus, a MT-dependent mechanism, leading to mild NE remnant compression by centrosomes (see *Figure 1C*), and a myosin II-dependent mechanism may work in concert to reduce CSV during this period.

A straightforward interpretation of these results is that myosin II within the actin network causes its contraction and the CSV reduction after NEBD. To obtain more evidence for this model, we performed light-induced local inhibition of myosin II activity in mitotic cells. We used azido-blebbistatin (azBB), which covalently binds the heavy chain of myosin II and inhibits its ATPase activity when

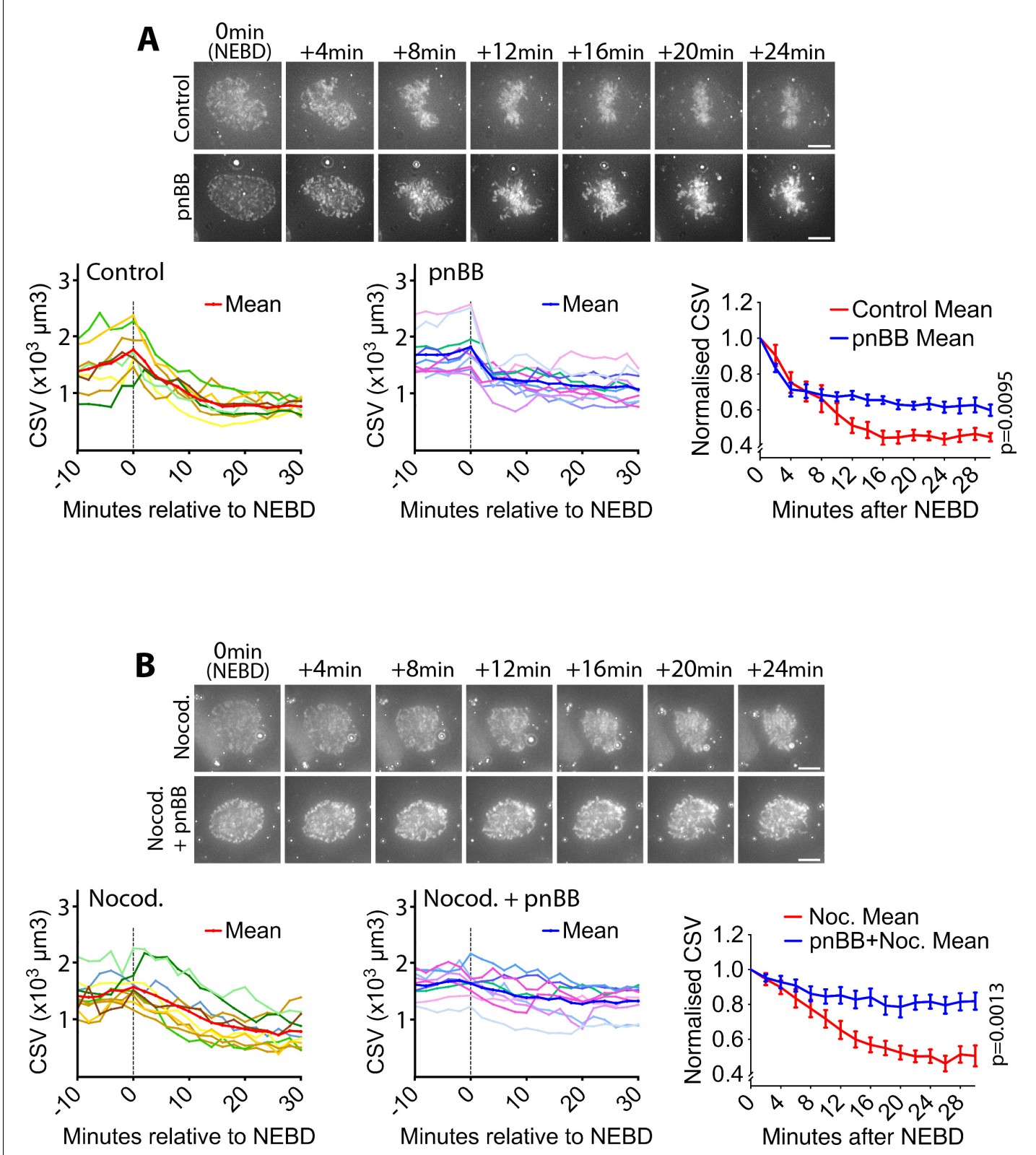

**Figure 3.** Myosin II activity is required for the CSV reduction after NEBD. (**A**) Inhibition of myosin II activity alleviates CSV reduction after NEBD. Images (z-projections) show representative cells with SiR-DNA-stained chromosomes, which entered mitosis in the presence of 50 μM paranitroblebbistatin (pnBB, bottom) or DMSO (control, top). Time was relative to NEBD. Timing of NEBD was determined by observation that GFP-LacI-NLS spread out of the nucleus (**Figure 1—figure supplement 1**). Scale bars, 10 μm. Left-hand and center graphs show CSV measurements in individual control and pnBB-

*Figure 3 continued on next page*

*Figure 3 continued*

treated cells, respectively (pnBB, n = 10; control, n = 8). The right-hand graph compares the means of normalized CSV (as in **Figure 1B**) in each condition. Error bars = s.e.m. *p* value (control vs pnBB) was obtained by two-way ANOVA. (**B**) CSV reduction is almost completely alleviated when myosin II activity is inhibited in the absence of MTs. Images (z-projections) show representative cells with SiR-DNA-stained chromosomes, which entered mitosis in the presence of 3.3 μM nocodazole plus 50 μm pnBB (Nocod. + pnBB, bottom), or 3.3 μM nocodazole (Nocod., top). 0.5 μM MK-1775 Wee1 inhibitor was used to allow cells to enter mitosis in both conditions. Time was relative to NEBD. Timing of NEBD was determined by observation that GFP-LacI-NLS spread out of the nucleus (**Figure 1—figure supplement 1**). Scale bars, 10 μm. Left-hand and center graphs show CSV measurements in individual cells treated by nocodazole alone and by nocodazole plus pnBB, respectively (nocodazole treated n = 10, nocodazole and pnBB treated n = 9). The right-hand graph compares the means of normalized CSV (as in **Figure 1B**) in each condition. Error bars = s.e.m. p-Value (nocodazole vs pnBB +Nocodazole) was obtained by two-way ANOVA.

DOI: https://doi.org/10.7554/eLife.46902.017

The following source data and figure supplements are available for figure 3:

**Source data 1.** Data at individual time points in individual cells.
DOI: https://doi.org/10.7554/eLife.46902.024
**Figure supplement 1.** Actin depolymerization inhibitor does not change CSV reduction after NEBD.
DOI: https://doi.org/10.7554/eLife.46902.018
**Figure supplement 2.** Myosin II co-localizes with the actin network on the NE in prophase.
DOI: https://doi.org/10.7554/eLife.46902.019
**Figure supplement 3.** Myosin II inhibition alleviates contraction of the actin network on the NE remnant after NEBD.
DOI: https://doi.org/10.7554/eLife.46902.020
**Figure supplement 4.** Myosin II inhibition does not affect chromosome mass volume (CMV) after NEBD.
DOI: https://doi.org/10.7554/eLife.46902.021
**Figure supplement 5.** Myosin II inhibition often leaves the actin network extending beyond the spindle poles soon after NEBD.
DOI: https://doi.org/10.7554/eLife.46902.022
**Figure supplement 6.** Atypical spindle morphology is found in LINC-DN-expressing and control cells, soon after NEBD.
DOI: https://doi.org/10.7554/eLife.46902.023

exposed to infra-red light (**Képiró et al., 2012**). A low concentration of azBB was used to ensure that myosin II was inhibited only at the region exposed to infra-red light. We exposed a half of the nucleus to infra-red light (**Figure 4A**) and compared the CSV over time in the exposed and non-exposed halves of the nucleus. In the exposed half of the nucleus (**Figure 4B,C**; blue), the CSV reduction was alleviated, compared to the non-exposed half (red). In the cells not treated with azBB (control), there was no significant difference in CSV between exposed (**Figure 4B,C**; green) and non-exposed (orange) halves of nuclei. These results support a model in which myosin II activity within the actin network (hereafter referred to as the 'acto-myosin' network) on NE remnants causes its contraction and CSV reduction after NEBD.

## Evidence that the CSV reduction in prometaphase by acto-myosin network contraction ensures timely congression and correct segregation of chromosomes

Following NEBD, spindle MTs interact with chromosomes while CSV is reduced by contraction of the acto-myosin network associated with NE remnants (**Figure 1B**). The acto-myosin network contraction might facilitate correct interaction between chromosomes and spindle MTs, leading to subsequent chromosome congression to the middle of the mitotic spindle. To test this, we measured the period between NEBD and completion of chromosome congression in cells expressing either LINC-DN or its control. LINC-DN expression led to a considerable delay in chromosome congression and anaphase onset, compared with the control (**Figure 5A–C**; **Video 7**), which was not associated with

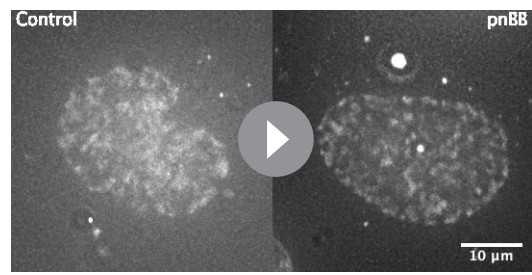

**Video 5.** Inhibition of myosin II activity alleviates CSV reduction after NEBD. Video of representative cells (Control and pnBB treated) shown in **Figure 3A** (SiR-DNA staining). Frame intervals are 2 min. Video is shown at 600x speed.
DOI: https://doi.org/10.7554/eLife.46902.025

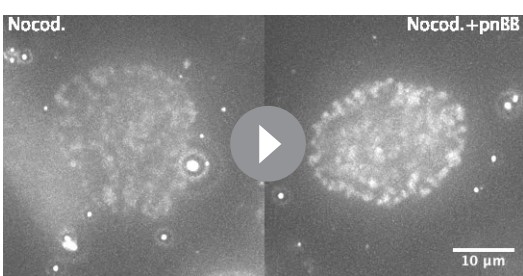

**Video 6.** CSV reduction is almost completely alleviated when myosin II activity is inhibited in the absence of MTs. Video of representative cells (Nocodazole and Nocodazole +pnBB treated) shown in *Figure 3B* (SiR-DNA staining). Frame intervals are 2 min. Video is shown at 600x speed.
DOI: https://doi.org/10.7554/eLife.46902.026

significant defects in spindle positioning or length (*Figure 5—figure supplement 1*). In cells expressing LINC-DN, chromosomes often remained behind a spindle pole and outside of the pole-to-pole region after NEBD (back-of-spindle chromosomes) (*Figure 5A* bottom and D, magenta arrowheads), which is consistent with the extension of NE remnants beyond spindle poles when contraction of the acto-myosin network was suppressed (*Figure 3—figure supplement 5*). In a subgroup of cells expressing LINC-DN where back-of-spindle chromosomes still remained at 20 min after NEBD in cells, there was a larger delay in congression (*Figure 5—figure supplement 2*). Moreover, LINC-DN expression led to more frequent abnormal segregation of chromosomes during anaphase (*Figure 5E*).

The above results with LINC-DN expression were reproduced with pnBB treatment (which inhibits myosin II) (*Figure 5—figure supplement 3*). Collectively, these results suggest that the CSV reduction due to contraction of the acto-myosin network diminishes back-of-spindle chromosomes and facilitates correct chromosome interaction with spindle MTs. These effects likely lead to timely chromosome congression and anaphase onset as well as high-fidelity chromosome segregation.

The above studies used human U2OS (near triploid) cells. We also investigated the acto-myosin network on the NE in other human cell lines. RPE cells (diploid) showed accumulation of actin on the NE in early mitosis (*Figure 5—figure supplement 4*, top). By contrast, HeLa cells (highly aneuploid) did not show such NE-associated actin accumulation (*Figure 5—figure supplement 4*, bottom). An interesting possibility is that a lack of the acto-myosin network is correlated with aneuploidy.

## Discussion

Our study has discovered a novel mechanism regulating chromosome positions, independently of spindle MTs, during early mitosis (*Figure 6*). In prophase, the acto-myosin network is formed on the cytoplasmic side of the NE, relying on the LINC complex. Following NEBD, the activity of myosin-II in the acto-myosin network promotes contraction of this network on the NE remnants. This contraction reduces CSV following NEBD, which in turn seems to facilitate chromosome alignment on the metaphase spindle (congression) and to ensure high-fidelity chromosome segregation.

We found that, with LINC-DN expression and pnBB treatment, chromosome congression to the middle of the mitotic spindle was significantly delayed (*Figure 5B* and *Figure 5—figure supplement 3*). To achieve efficient chromosome congression, chromosomes must promptly establish initial interaction with spindle MTs and subsequently move from a position near the spindle poles to a position at the middle of the spindle. A delay in either step would lead to inefficient congression – which step, then, was delayed with LINC-DN expression and pnBB treatment (and with consequent defects in CSV reduction)? We presume that LINC-DN expression and pnBB treatment mainly delayed chromosome movement to the middle of the spindle, rather than the initial interaction. Consistent with this notion, LINC-DN expression and pnBB treatment diminished CSV reduction during 8–16 min after NEBD (*Figures 2D* and *3A*); however at this stage, the initial chromosome–MT interaction had likely already occurred – for example, during 6–8 min after NEBD, many chromosomes had moved inward away from the NE remnants, probably due to MT interaction (*Figure 2—figure supplement 2B,C*). Moreover, the appearance of back-of-spindle chromosomes after LINC-DN expression was well correlated with a delay in chromosome congression (*Figure 5—figure supplement 2*). We reason that, during 8–16 min after NEBD, the contractile acto-myosin network constrained chromosomes within the pole-to-pole region (*Figure 3—figure supplement 5*), thus reducing the occurrence of back-of-spindle chromosomes (*Figure 5D* and *Figure 5—figure supplement 3D*) and facilitating chromosome movement to the middle of the spindle, that is congression. However, the CSV reduction also occurs during an earlier phase (0–8 min after NEBD), and it is very possible that

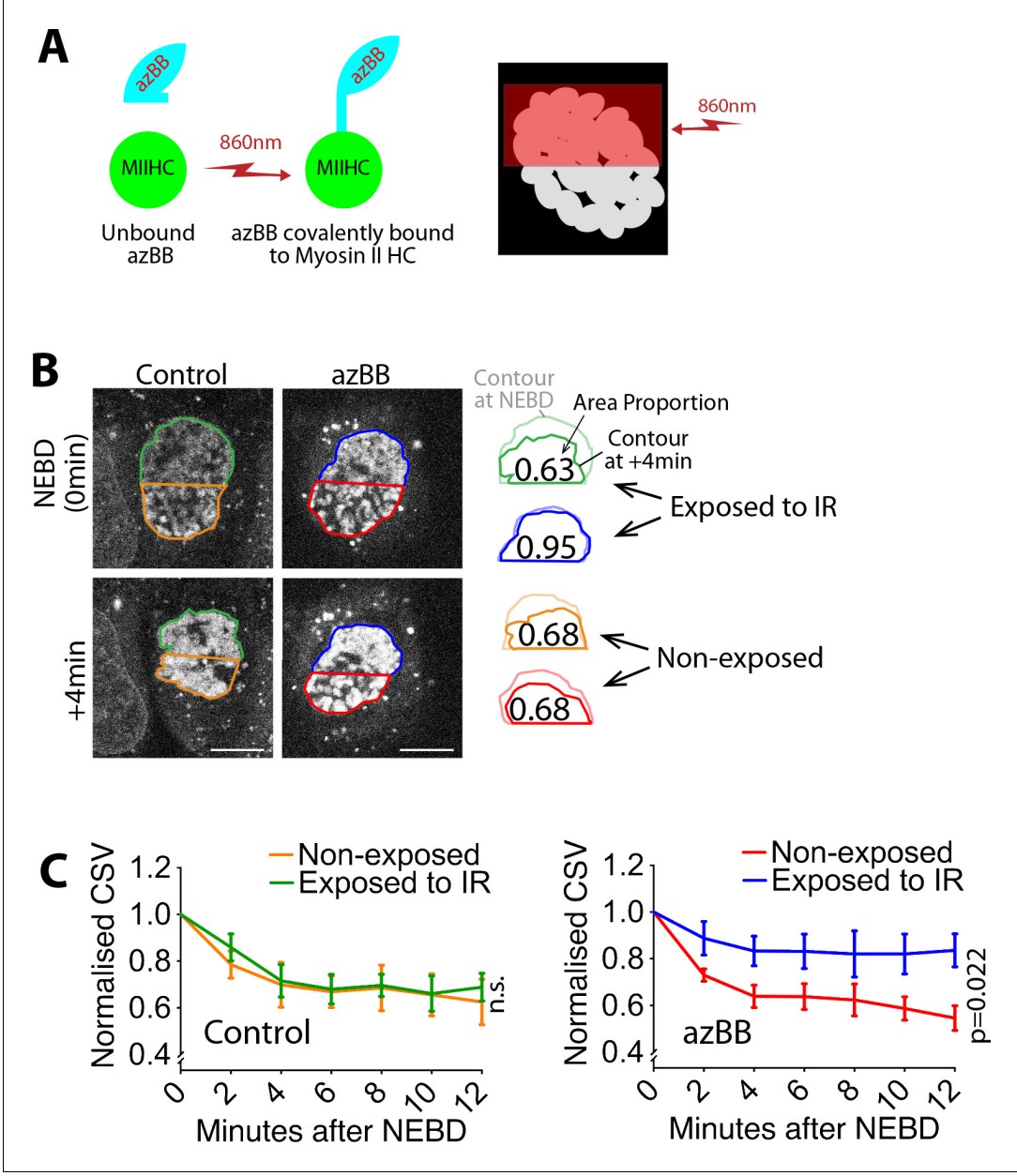

**Figure 4.** Local inhibition of myosin II activity results in local alleviation of CSV reduction. (**A**) Diagram shows that azidoblebbistatin (azBB) is covalently linked to the myosin II heavy chain and inhibits its activity in the half of the nucleus exposed to infrared light. (**B**) Images (projections of three z-sections) show SiR-DNA-stained chromosomes in representative cells that were incubated in the presence of 5 μM azidoblebbistatin (azBB, right) or DMSO (control, left). The half of the nucleus in blue and green was exposed to 860 nm light just prior to NEBD, while the half in red and orange was not. Time was relative to NEBD. Timing of NEBD was determined by observation that GFP-LacI-NLS spread out of the nucleus (*Figure 1—figure supplement 1*). Scale bars, 10 μm. Diagrams show how the indicated area was reduced at +4 min, relative to the area at NEBD. (**C**) Graphs show means of normalized CSV in half of the nucleus, which was exposed vs non-exposed to infra-red (IR), in the presence (bottom) and absence (top) of azBB (control n = 5, azBB n = 7). Colors of lines match the colors that border the half nuclei in D. p-Value (exposed vs non-exposed to IR) was obtained by two-way ANOVA. n.s., not significant. Error bars, s.e.m. CSV was normalized as in *Figure 1B*.

DOI: https://doi.org/10.7554/eLife.46902.027

The following source data is available for figure 4:

**Source data 1.** Data at individual time points in individual cells.

DOI: https://doi.org/10.7554/eLife.46902.028

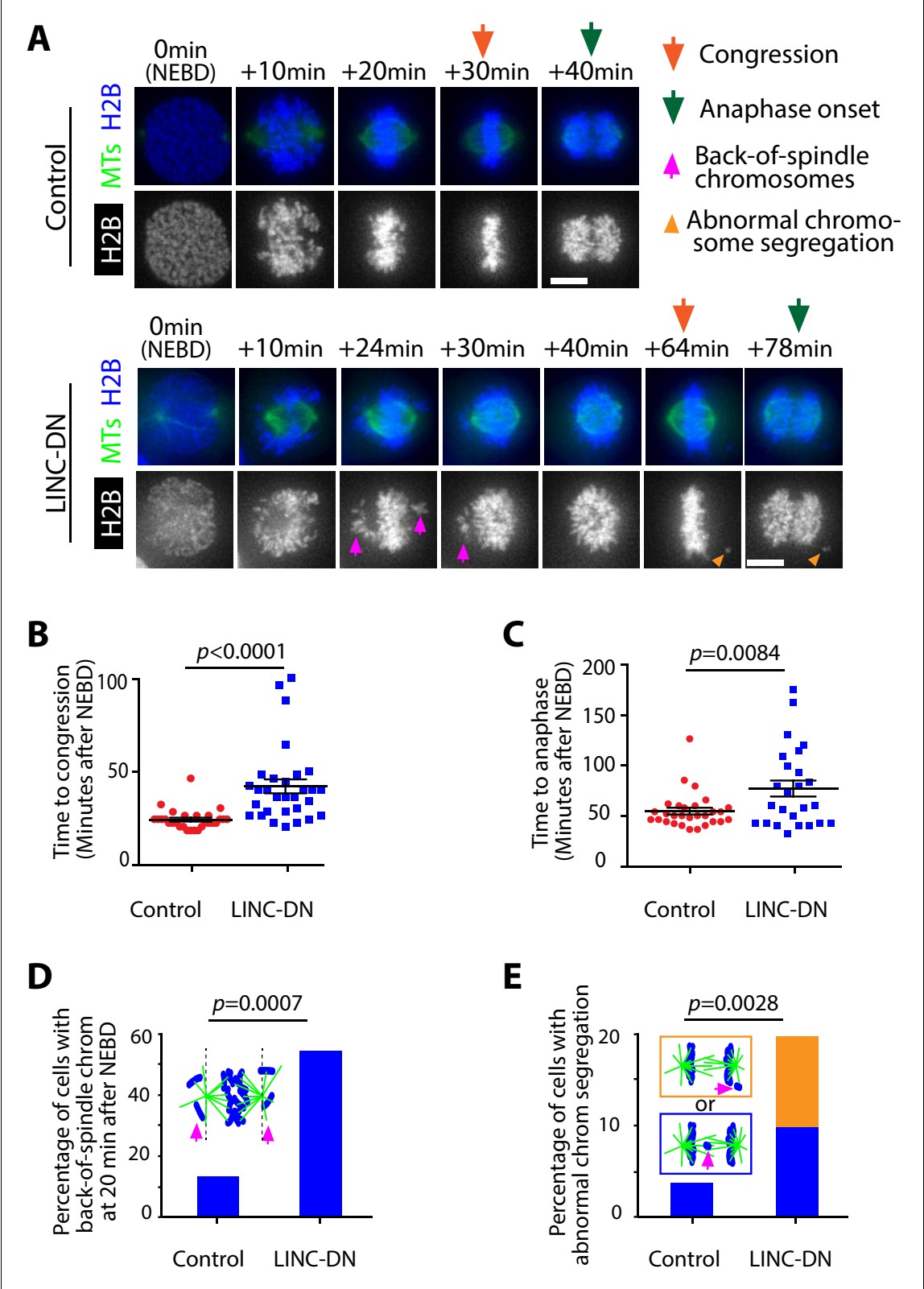

**Figure 5.** Evidence that removal of actin network delays chromosome congression and anaphase onset, and increases frequency of abnormal chromosome segregation. (A) Images (z-projections) show representative cells expressing histone 2B (H2B)-cerulean and GFP-tubulin. They progress from NEBD to anaphase, expressing either LINC-DN or a control construct. Orange arrows indicate time points at which congression was completed, while green arrows indicate time points of anaphase onset. Magenta arrows indicate chromosomes at backside of the spindle (back-of spindle

*Figure 5 continued on next page*

*Figure 5 continued*

chromosomes; see D). Light orange arrowheads indicate abnormal chromosome segregation during anaphase (see E). (B) Graph compares time (minutes) from NEBD to chromosome congression in cells expressing LINC-DN (n = 30) versus a control construct (n = 28). p-Value was obtained by *t*-test, error bars = s.e.m. (C) Graph compares time (minutes) from NEBD to the anaphase onset in cells expressing LINC-DN (n = 25) versus a control construct (n = 29). p-Value was obtained by *t*-test, error bars = s.e.m. (D) Graph shows the percentage of cells with chromosomes at backside of the spindle (back-of-spindle chromosomes) at 20 min after NEBD in cells expressing LINC-DN (n = 37) versus a control construct (n = 31). *p* value was obtained by Fisher's exact test. (E) Graph shows the percentage of cells with abnormal chromosome segregation during anaphase, in cells expressing LINC-DN (n = 41) versus a control construct (n = 83). Two types of abnormal chromosome segregation were observed in LINC-DN-expressing cells during anaphase, that is chromosomes at the back-of-spindle (orange) and lagging chromosomes between two poles (blue), which were color-coded in diagram and separately counted in graph. p-Value was obtained by Fisher's exact test (three categories; normal segregation and two types of abnormal segregation).

DOI: https://doi.org/10.7554/eLife.46902.029

The following source data and figure supplements are available for figure 5:

**Source data 1.** Data in individual cells.
DOI: https://doi.org/10.7554/eLife.46902.034

**Figure supplement 1.** LINC-DN expression did not change the distance between centrosomes and did not cause deviation of the spindle from the center of the nucleus.
DOI: https://doi.org/10.7554/eLife.46902.030

**Figure supplement 2.** The presence of back-of-spindle chromosomes soon after NEBD leads to a delay in chromosome congression.
DOI: https://doi.org/10.7554/eLife.46902.031

**Figure supplement 3.** pnBB treatment delays chromosome congression and anaphase onset, and increases frequency of abnormal chromosome segregation.
DOI: https://doi.org/10.7554/eLife.46902.032

**Figure supplement 4.** RPE cells show accumulation of actin on the NE in early mitosis, but HeLa cells do not.
DOI: https://doi.org/10.7554/eLife.46902.033

the earlier CSV reduction facilitates the initial chromosome–MT interaction. While this possibility is intriguing, we currently lack the ability to inhibit the CSV reduction during this phase and to analyze consequent chromosome–MT interactions.

Our result suggests that contraction of the acto-myosin network on NE remnants starts immediately after NEBD to facilitate CSV reduction (*Figure 2—figure supplement 1* and *Figure 1B*). What triggers the contraction of the acto-myosin network at this specific timing in the cell cycle? We speculate that NEBD reduces tension on the NE, which may trigger the contraction of the acto-myosin network on the NE remnants. Similar mechanisms have been reported (*Pontes et al., 2017*); for example, during cell migration, reduced membrane tension within lamellipodia led to local contraction of the acto-myosin network (*Gauthier et al., 2011*).

Recent studies have revealed important roles of the actin cytoskeleton and non-muscle myosin in high-fidelity chromosome segregation. For example, in starfish oocytes, actin depolymerization beneath the NE remnants facilitates the interaction of chromosomes with spindle MTs (*Burdyniuk et al., 2018*; *Lénárt et al., 2005*). In *Xenopus* embryos, spindle integrity relies on actin on the spindle and myosin-10 at spindle poles (*Woolner et al., 2008*). In PtK2 cells, actin and non-muscle myosin on the mitotic spindle regulate the length of the spindle (*Sheykhani et al., 2013*). In mouse oocytes, actin dynamics on the spindle promote robust kinetochore–MT interactions (*Mogessie and Schuh, 2017*). Moreover, several studies suggest that actin and myosin-II on the cell cortex are important for formation and correct orientation of the mitotic spindle (*Rosenblatt et al., 2004*; *Whitehead et al., 1996*; *Kunda and Baum, 2009*). In the current study, we found a novel actin- and myosin II-dependent mechanism

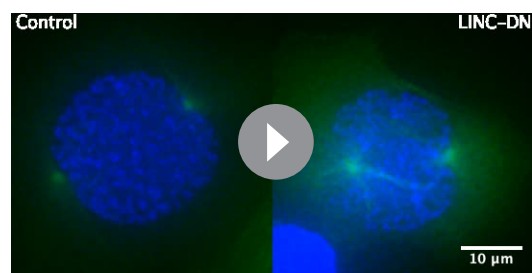

**Video 7.** Evidence that removal of actin network delays chromosome congression and anaphase onset. Video of representative cells (Control and LINC-DN expressing) shown in *Figure 5A* (expressing GFP-tubulin and H2B-cerulean). Frame intervals are 2 min. Video is shown at 600x speed.
DOI: https://doi.org/10.7554/eLife.46902.035

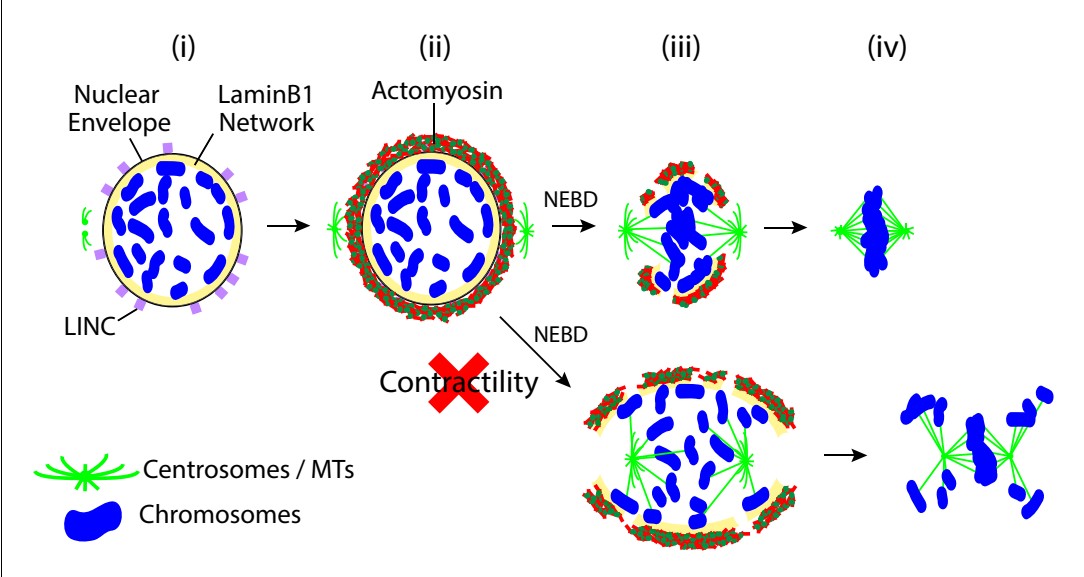

**Figure 6.** Summary diagram. Prior to NEBD, the LINC complex promotes the accumulation of acto-myosin to the cytoplasmic side of the nuclear envelope (**i, ii**). Upon NEBD, myosin II promotes contraction of the acto-myosin network on the NE remnant, thus reducing the CSV (**iii**). Constraining chromosomes from scattering results in efficient chromosome interaction with the spindle MTs and timely chromosome congression (**iii, iv**).
DOI: https://doi.org/10.7554/eLife.46902.036

promoting efficient and correct interaction between chromosomes and the mitotic spindle in human cells. In contrast to the mechanisms mentioned above, this mechanism is dependent on the acto-myosin network formed on the cytoplasmic surface of the NE and relies on its contraction driven by myosin II. Thus, the actin cytoskeleton and non-muscle myosin appear to contribute to high-fidelity chromosome segregation through multiple mechanisms. It will be intriguing to explore how these mechanisms are conserved in evolution and used in different cell types to ensure correct chromosome segregation.

# Materials and methods

**Key resources table**

| Reagent type (species) or resource | Designation | Source or reference | Identi-fiers | Additional information |
|---|---|---|---|---|
| Cell line (Human) | U2OS *cdk1-as* | *Rata et al., 2018* | | |
| Cell line (Human) | wild-type U2OS | ATCC | HTB-96 | |
| Cell line (Human) | HeLa | ATCC | CCL-2 | |
| Cell line (Human) | hTERT RPE | ATCC | CRL-4000 | |
| Transfected construct (Human) | p3'SSclonedimer pF9-poly-EGFP-lacI-NLS (GFP-LacI-NLS) | *Hori et al., 2013* | | |
| Transfected construct (Human) | mCherry-Lifeact-7 (mCherry-Lifeact) | Addgene | #54491 | A gift from Michael Davidson |
| Transfected construct (Human) | pGFP-alphaTub (GFP-Tubulin) | Other | | A gift from Jason Swedlow lab |

*Continued on next page*

*Continued*

| Reagent type (species) or resource | Designation | Source or reference | Identi-fiers | Additional information |
|---|---|---|---|---|
| Transfected construct (Human) | mRFP1-SR-KASH (LINC-DN) | *Luxton et al., 2010* | | |
| Transfected construct (Human) | mRFP1-KASH-ΔL (Control) | *Luxton et al., 2010* | | |
| Transfected construct (Human) | pCS-H2B-cerulean (H2B-Cerulean) | Other | | A gift from Sean Megason Lab |
| Transfected construct (Human) | CMV-GFP-NMHC II-A (MHC-GFP) | AddGene | #11347 | A gift from Robert Adelstein |
| Antibody | anti-Tubulin (Rat monoclonal) | Cell Signalling Technology | YL1/2 | Used at 1:1000 dilution |
| Antibody | anti-LaminB1 (Rabbit polyclonal) | Abcam | ab16048 | Used at 1:1000 dilution |
| Antibody | anti-Actin (Rabbit polyclonal) | Thermo-Fisher | PA1-183 | Used at 1:1000 dilution |
| Antibody | A488 goat anti-Rabbit | Thermo-Fisher | A-11034 | Used at 1:1000 dilution |
| Antibody | A488 donkey anti-Rat | Thermo-Fisher | A-21208 | Used at 1:1000 dilution |
| Antibody | A594 goat anti-Rabbit | Thermo-Fisher | A-11037 | Used at 1:1000 dilution |
| Chemical compound, drug | Phalloidin DyLight 650 | Invitrogen | 13454309 | Used at 1:1000 dilution |
| Chemical compound, drug | Hoechst 33342 | Sigma-Aldrich | 14533 | Used at 1:2000 dilution |
| Chemical compound, drug | SiR-DNA | tebu-bio | SC007 | Used at 100 nm |
| Chemical compound, drug | 1NM-PP1 | Merck Millipore | 529581 | Used at 1 µM |
| Chemical compound, drug | RO-3306 | Merck Millipore | 217699 | Used at 10 µM |
| Chemical compound, drug | nocodazole | Sigma | M1404 | Used at 3.3 µM |
| Chemical compound, drug | paranitrob lebbistatin (pnBB) | motorpharmacology | | Used at 50 µM |
| Chemical compound, drug | azidoblebbistatin (azBB) | motorpharmacology | | Used at 5 µM |
| Chemical compound, drug | jasplakinolide | Merck | J4580 | Used at 1 µM |
| Chemical compound, drug | MK-1775 | Selleck Chemicals | S1525 | Used at 500 nM |
| Software, algorithm | Imaris, 3D surface tool | Bitplane | | |
| Software, algorithm | Imaris, Convex Hull generation tool | Bitplane | | Convex Hull tool from filament tool adapted for 3D surface tool |
| Software, algorithm | Fiji | https://fiji.sc/ | | |

## Cell culture

To synchronize U2OS cells in early mitosis with a chemical genetic approach, their *CDK1* genes were replaced with *cdk1-as* construct (*Rata et al., 2018*), as previously done in DT40 cells

(*Hochegger et al., 2007*). The U2OS *cdk1-as* cells, HeLa cells (ATCC CCL-2) and original U2OS cells (ATCC HTB-96) were cultured in DMEM (Gibco 41965–039) medium supplemented with 10% FBS (Gibco 10500) and 1% 10,000 U/ml Pen/strep (Gibco 15140–122). hTERT RPE (ATCC CRL-4000) were maintained in DMEM-F12 medium (Gibco 11330–057) supplemented with 10% FBS (Gibco 10500) and 1% 10,000 U/ml Pen/strep (Gibco 15140–122). To detach cells from culture flasks or dishes, they were treated with 0.05% Trypsin (Gibco 25300–054). All cell lines used in this study were verified by Eurofins authentication service, using STR profiling. There was no contamination of mycoplasma in cell cultures in tests using Mycoalert mycoplasma detection kits (Lonza LT07-118).

## Expression constructs and transfection

Plasmid DNA constructs used in this study were: GFP-LacI-NLS (*Hori et al., 2013*; pT2699), mCherry-Lifeact (a gift from Michael Davidson, Addgene #54491; pT3188), GFP-Tubulin (a gift from Jason Swedlow lab; pT2932), LINC-DN (mRFP1-SR-KASH; *Luxton et al., 2010*; pT3138) and LINC-DN Control (mRFP1-KASH-ΔL; *Luxton et al., 2010*; pT3139); H2B-Cerulean (a gift from Sean Megason lab; pT2698), and MHC-GFP (CMV-GFP-NMHC II-A, a gift from Robert Adelstein, AddGene #11347; pT3153). For transfection of plasmids, reactions were prepared in 135 µl OptiMem medium plus Glutamax (Gibco 51985–026) with 3 µl DNA FuGene HD Transfection Reagent (Promega E2311) for every 1 µg of plasmid DNA.

## Small molecule inhibitors

Small molecule inhibitors were used with the following concentrations: 1NM-PP1 1 µM (Merck Millipore 529581), RO-3306 10 µM (Merck Millipore 217699), nocodazole 3.3 µM (Sigma M1404), paranitroblebbistatin (pnBB) 50 µM (motorpharmacology), azidoblebbistatin (azBB) 5 µM (motorpharmacology), jasplakinolide 1 µM (Merck J4580) and MK-1775 500 nM (Selleck Chemicals S1525). A Wee1 inhibitor MK-1775 was used to facilitate entry into mitosis by bypassing G2/M checkpoint (*Hirai et al., 2009*) in our experiments with nocodazole (*Figures 1C* and *3B*) and with LINC-DN plus GFP-LacI-NLS (*Figure 2D*). Use of MK-1775 alone did not significantly change kinetics of CSV reduction – compare *Figure 2D*, red (with MK-1775) and *Figure 3A*, red (without MK-1775).

## Microscopy

Live-cell images and immunofluorescence images were acquired using a Deltavision Elite widefield microscopy system (GE Healthcare) with 100x and 60x objective lenses (Olympus, NA 1.4) and cameras Cascade 1K EMCCD (Roper Scientific) and CoolsnapHQ2 (Photometrics). For live-cell imaging, the microscope chamber was maintained at 37°C with 5% carbon dioxide. For photoactivation of azBB, we used Zeiss 710 confocal microscopy system with a Coherent Chameleon multiphoton laser attachment and with 63x objective lens (NA 1.4). Super-resolution images were acquired using a Zeiss Airyscan 880 system with 63x objective lens (NA 1.4). Widefield Images were deconvolved using SoftWorx and images were analyzed using Fiji, OMERO, Imaris and Volocity software.

## Live-cell imaging

U2OS *cdk1-as* cells were synchronized and prepared for live-cell imaging as follows: cells were plated out at 50–60% confluency onto glass-bottomed imaging dishes (FluoroDish; World Precision Instruments, FD35-100). The following evening they were transfected with the relevant plasmids after replacement of the media with media lacking pen/strep. The transfection was allowed to continue overnight before the transfection reagents were washed out with further media without pen/strep. Cells were then arrested in the evening by addition of 1NM-PP1 (in the case of U2OS *cdk1-as* cells) or RO-3306 (in the case of HeLa or RPE cells), alongside SiR-DNA to stain chromatin where required. At this stage, media was also replaced with Fluorobrite medium (Gibco A18967-01) supplemented with 10% FBS (Gibco 10500), 2 mM L-Glutamine (Gibco 200 mM, 25030–025), 25 mM HEPES (1 m Lonza BE17-737E) and 1 mM Na-Pyruvate (100 mM, Lonza BE13-115E). The following day cells were released from arrest by washing in 2 ml Fluorobrite medium (with supplements) 12 times, immediately followed by image acquisition. pnBB, azBB and MK-1775 were added after the final wash, prior to imaging. Nocodazole was added 1 hr before release, and added again after the final wash, prior to imaging. GFP-Tubulin construct was stably transfected in cells shown in *Figure 2A*, *Figure 2—figure supplement 2*, *Figure 5A*, *Figure 3—figure supplements 3*, *5* and

*6*, and *Figure 5—figure supplement 3*. Images were acquired every 2 min except for a) *Figure 2A*, *Figure 2—figure supplement 2* and *Figure 3—figure supplement 5* where images were acquired every minute, and b) *Figure 1—figure supplement 2* where images were acquired every 5 min. During imaging, 10–12 z-sections were acquired with 1.6 μm z-intervals, except for a) *Figure 2A*, *Figure 1—figure supplement 3*, *Figure 3—figure supplement 2* and *Figure 5—figure supplement 4*, where 10–12 z-sections were acquired 0.8 μm z-intervals and b) *Figure 2—figure supplement 2* and *Figure 3—figure supplement 5*, where 5 z-sections were acquired with 1.6 μm z-intervals. For live-cell imaging of asynchronous U2OS cells, they were treated in the same way but were not arrested overnight before imaging. When a cell drifted during live-cell imaging, the frame of time sequence images in figures was adjusted so that a chromosome mass was still positioned at the center of the frame in each image. The timing of NEBD was determined by (a) the efflux of GFP-LacI-NLS signals from the nucleus into the cytoplasm, (b) the influx of GFP tubulin signals from the cytoplasm into the nucleus, or (c) the influx of GFP-NMHCII-A signals from the cytoplasm into the nucleus.

## Cell preparation for immunofluorescence

The night before fixation, media was replaced with Fluorobrite medium with supplements and the cells arrested with 1NM-PP1 if required. On the following day, cells were released by washing in 2 mL Fluorobrite medium 12 times. Progression into mitosis was confirmed under a phase contrast light microscope. Then cells were fixed when they reached an appropriate stage, based on observation of chromosome condensation. To fix cells for immuno-staining, they were rinsed with pre-warmed PBS and incubated for 10 min with 37°C 4% methanol-free formaldehyde at 37°C. They were then rinsed three times with PBS and permeabilized for 10 min in PBS with 0.5% Triton X-100. Following this they were blocked with 5% BSA in PBS for an hour at 37 °C. While blocking, primary antibodies were prepared in 5% BSA in PBS to the relevant dilution. Following blocking, cells were incubated rocking with the primary antibody mixture for at least 60 mins at room temperature (or 4° C if incubated overnight). Following primary incubation, cells were washed rocking in PBS for 5 min, three times. Meanwhile, secondary antibodies plus phalloidin/Hoechst 33342 were prepared to the relevant dilution in 5% BSA in PBS. Cells were then incubated rocking with secondary antibodies for at least 60 min at room temperature (or 4 °C if incubated overnight). Following secondary incubation, cells were washed rocking in PBS for 5 min, three times.

## Immunofluorescence and cell staining

Primary antibodies were used for immuno-staining with the following dilution: anti-Tubulin 1:1000 (Cell Signalling Technology YL1/2), anti-LaminB1 1:1000 (Abcam ab16048) and anti-Actin 1:1000 (Thermo-Fisher PA1-183). Secondary antibodies were used with the following dilution: A488 goat anti-Rabbit 1:1000 (Thermo-Fisher A-11034), A488 donkey anti-Rat 1:1000 (Thermo-Fisher A-21208), and A594 goat anti-Rabbit 1:1000 (Thermo-Fisher A-11037). Dyes were used for cell staining with the following dilution: Phalloidin DyLight 650 1:1000 (Invitrogen 13454309), Hoechst 33342 1:2000 (Sigma-Aldrich 14533), SiR-DNA 100 nm (tebu-bio SC007).

## Azidoblebbistatin photoactivation experiments

U2OS *cdk1-as* cells expressing GFP-LacI-NLS were prepared for live-cell imaging as described above. SiR-DNA was supplemented when 1NM-PP1 was added to arrest cells. Azidoblebbistatin (azBB) was added when cells were released from the arrest (by washing off 1NM-PP1). To identify cells in prophase, the start of chromosome condensation was monitored with SiR-DNA signal. A scan area was zoomed in so that it covered roughly half of the nucleus in prophase. A z-stack (three sections, 2 μm apart) of the region was then rapidly scanned four times with 633 nm and 488 nm lasers to observe chromosomes and GFP-LacI-NLS, respectively, alongside 860 nm multi-photon laser exposure to photoactivate the covalent binding of azBB to Myosin II. The region was scanned in three rounds in this way at 1-min interval. The scan area was then extended so that image of the entire nucleus could be acquired. The whole nucleus was then imaged for SiR-DNA and GFP-LacI-NLS in a z-stack of 20 μm with 2 μm z-interval, every 2 min. The timing of NEBD was judged based on the efflux of GFP-LacI-NLS from the nucleus into the cytoplasm. The nuclear region exposed to 860 nm light could be discriminated based on partial photo-bleaching of SiR-DNA signals.

## Measurement of chromosome scattering volume

U2OS *cdk1-as* cells expressing GFP-LacI-NLS were imaged every 2 min in the presence of SiR-DNA. Z-stacks of 10-sections were acquired with 2 µm interval. The CSV was measured as follows: The surface of chromosomes was identified using Imaris (3D surface object tool). Obviously non-chromosome objects were excluded in this process. On each z-section, the surface contours of multiple chromosomes were connected into one single-stroke contour (*Figure 1B*, green lines imposed on images), using Imaris, in such a way that the connection doesn't affect 2D convex hull. These contours were used to generate a continuous 3D surface contour, which was required to compute a 3D convex hull. Based on this continuous 3D surface contour, a 3D convex hull was generated using Imaris (convex hull generation tool). The volume of this 3D convex hull was defined as the CSV. To compare reduction of CSV between two conditions, we used two-way ANOVA in which data at all the time points shown in relevant graphs were included in analyses.

## Measurement of the volume and intensity of the actin network

U2OS *cdk1-as* cells expressing mCherry-Lifeact and GFP-tubulin were imaged every 2 min. Z-stacks of 10-sections were acquired at 2 µm intervals. A 3D surface object was generated using Imaris by creating contours following the actin network at each z-section. The volume of the surface object was taken as the volume inside the actin network. Meanwhile, to measure the intensity of actin network, a two pixel wide line was drawn along the actin network on a single z-section using Fiji; the shape of the nucleus as seen by the exclusion of GFP-tubulin background was used as a guide when the actin signals were ambiguous. The mean gray value of the pixels within this line was taken as a measurement. As a background reading, a measurement was taken from a similar two-pixel wide line drawn in the cytoplasmic region surrounding the nucleus. To generate a final reading, the measurement from the cytoplasm was subtracted from the measurement from the actin network.

## Measurement of chromosome mass volume

U2OS *cdk1-as* cells imaged every 2 min in the presence of SiR-DNA. Z-stacks of 10 sections were acquired with 2 µm interval. A 3D surface covering SiR-DNA signal was automatically generated with Imaris as follows; the mean value in the Cy5 channel was set as the threshold, and the Background Subtraction (local contrast) option was chosen. The volume of the surface object was defined as the chromosome mass volume (CMV).

## Timing of chromosome congression and anaphase onset

U2OS *cdk1-as* cells expressing GFP-tubulin and H2B-Cerulean were imaged every 2 min. Z-stacks of 10-sections were acquired with 2 µm interval. Timing of complete congression was estimated as the time all chromosomes aligned on the spindle equator. Timing of anaphase onset was estimated as the time chromosomes began segregation to opposite poles.

## Replicates and statistics

All experiments were repeated at least twice and similar results were obtained. Statistical analyses were carried out using Prism software (Graphpad). Methods of statistics are stated in each relevant figure legend. All p-values were two-tailed.

## Acknowledgements

We thank Tanaka lab members, A Ciulli, A Testa, J Januschke and M Gierlinski for discussion; G Ball for help with image analysis; L Clayton for editing the manuscript;, T Fukagawa, J Swedlow, S Megason, R Adelstein, M Davidson, and RY Tsien for reagents; S Swift, and P Appleton for microscope maintenance. This work was supported by the Wellcome Trust (096535/Z/11/Z, 097945/Z/11/Z, 208401/Z/17/Z), Cancer Research UK (C28206/A114499) and Medical Research Council (MR/K015869/1). TUT is a Wellcome Trust Principal Research Fellow. The authors declare no competing financial interests.

## Additional information

### Funding

| Funder | Grant reference number | Author |
|---|---|---|
| Wellcome | 096535/Z/11/Z | Tomoyuki Tanaka |
| Wellcome | 097945/Z/11/Z | Tomoyuki Tanaka |
| Wellcome | 208401/Z/17/Z | Tomoyuki Tanaka |
| Cancer Research UK | C28206/A114499 | Helfrid Hochegger |
| Medical Research Council | MR/K015869/1 | Tomoyuki Tanaka |

The funders had no role in study design, data collection and interpretation, or the decision to submit the work for publication.

### Author contributions

Alexander JR Booth, Conceptualization, Formal analysis, Investigation, Visualization, Methodology, Writing—original draft, Writing—review and editing; Zuojun Yue, Formal analysis, Investigation, Visualization, Methodology; John K Eykelenboom, Tom Stiff, Resources, Methodology; GW Gant Luxton, Resources, Methodology, Writing—original draft; Helfrid Hochegger, Resources, Supervision, Funding acquisition, Methodology, Writing—original draft; Tomoyuki U Tanaka, Conceptualization, Supervision, Funding acquisition, Visualization, Methodology, Writing—original draft, Project administration, Writing—review and editing

### Author ORCIDs

Alexander JR Booth (iD) http://orcid.org/0000-0002-3320-7919
John K Eykelenboom (iD) https://orcid.org/0000-0003-4115-9686
GW Gant Luxton (iD) http://orcid.org/0000-0002-6180-8906
Helfrid Hochegger (iD) https://orcid.org/0000-0001-8366-6198
Tomoyuki U Tanaka (iD) https://orcid.org/0000-0002-9886-5947

### Decision letter and Author response
Decision letter https://doi.org/10.7554/eLife.46902.039
Author response https://doi.org/10.7554/eLife.46902.040

## Additional files

### Supplementary files
• Transparent reporting form
DOI: https://doi.org/10.7554/eLife.46902.037

### Data availability
A source data file has been provided for each figure, and it contains the source data at individual time points in individual cells where relevant.

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
