## [Decision Letter]

Thank you for submitting your article "Contractile acto-myosin network on nuclear envelope remnants positions human chromosomes for mitosis" for consideration by *eLife*. Your article has been reviewed by three peer reviewers, one of whom is a member of our Board of Reviewing Editors, and the evaluation has been overseen by Anna Akhmanova as the Senior Editor. The following individual involved in review of your submission has agreed to reveal their identity: Helder Maiato (Reviewer #3).

The reviewers have discussed the reviews with one another and the Reviewing Editor has drafted this decision to help you prepare a revised submission.

This manuscript by Booth et al. describes an acto-myosin network that assembles on the cytoplasmic surface of the nuclear envelope prior to mitosis. This network seems to be present in some human cell lines (U2OS, RPE) but not others (HeLa). The authors provide evidence that contraction of the network after nuclear envelope breakdown gathers chromosomes in the spindle center, which may support chromosome attachment to the spindle. This is a novel observation that extends the repertoire of molecular mechanisms that act during early mitosis to establish proper chromosome attachment.

Summary:

Actin networks are known to influence mitosis and meiosis beyond their role in cytokinesis. An actin cortex underneath the plasma membrane supports mitotic cell rounding, spindle assembly and spindle positioning. Actin filaments can support spindle function, and nuclear actin has been shown to help expand the nucleus after mitosis. In starfish oocytes, chromosomes are collected towards the spindle by an actin network, and in mammalian eggs actin supports chromosome attachment to spindle microtubules.

The work reported in this manuscript expands this array of actin functions. The authors establish that the volume that chromosomes occupy (the 'chromosome scattering volume', CSV) decreases in the first ~8 minutes after nuclear envelope breakdown and continues to drop over the subsequent 20 minutes. They found that reduction of the CSV was impaired by a dominant-negative LINC complex mutation, which also abolished the pre-mitotic enrichment of actin around the nuclear envelope. Inhibition of myosin II activity similarly impaired CSV reduction, suggesting that it is driven by contraction of the actin network surrounding the nucleus. The effect was enhanced by additional depolymerization of microtubules, suggesting that microtubules and actin cooperate. The same dominant-negative LINC mutation also led to chromosome attachment defects, and the authors concluded that the reduction in CSV helps chromosome capture by the mitotic spindle.

Overall, this paper reports novel and interesting observations on the early stages of mitosis in human cells, which is an area with many unknowns, and the paper thus constitutes a significant advance for the field. The experiments are for the most part well designed, executed and controlled.

Essential revisions:

- The conclusion that the acto-myosin network contraction positions chromosomes for attachment to the spindle, as stated in the title, is not well substantiated. The authors do not directly assess chromosome attachment. The initial reduction of CSV (during the first ~8 min) remains intact in LINC-DN or pnBB-treated cells, and this is the time when chromosomes are expected to attach to the spindle. Unless the authors provide direct evidence, this conclusion needs to be toned down.

- An alternative explanation for the observed attachment defects is that microtubule-related mechanisms of congression are disrupted. Spindles in pnBB-treated cells clearly have an atypical morphology (Figure 3—figure supplement 3 and Figure 3—figure supplement 6) and it remains unclear whether this is secondary to attachment problems or could be the primary cause of the attachment defects. The authors should examine mitotic spindles in the LINC-DN and in jasplakinolide-treated cells to determine whether the spindle defects co-occur with the failure to reduce CSV. Additional visualization of a kinetochore marker would allow the authors to judge when chromosome attachment to the spindle happens in these cells.

- The text is very concise, which makes it easy to read. However, the reviewers felt that it was too succinct, and that the discussion of prior work was not given adequate space. Prior work on the position and movement of chromosomes in early mitosis, on the relation between actin/myosin and spindles and on the 'spindle envelope' seems relevant. Papers that were mentioned as important by the reviewers include: Baarlink et al., NCB 2017; Barisic et al., 2014; Itoh et al., 2018; Khodjakov work on chromosome congression; Kunda and Baum, 2009; Whitehead et al., 1996; Rosenblatt et al., 2004; Heng et al., J Cell Sci 2012; Woolner et al., 2008; papers by A. Forer (Protoplasma 2007, Cytoskeleton 2013,.…).

- In many experiments, the authors only analyzed a small number of cells, and it seems that control cells from different days yield average normalized CSV curves that are significantly different from each other. For the most part, the differences between control- and treated cells seem large enough that this is not a concern. However, the conclusion that nocodazole-treated cells differ at the 4-min time point from control cells (Figure 1C) does not seem robust. Picking different control cells would yield a different conclusion. The authors either need to image more cells or tone down their conclusion. The number of asynchronously growing cells observed for CSV at mitotic entry (Figure 1—figure supplements 1-4) should be increased beyond three.

- Since the authors speculate about potential segregation defects in the absence of the CSV-reduction mechanism, it would be appropriate to analyze chromosome segregation in the LINC-DN or pnBB cells, which is easily feasible.

---

## [Author Response]

Essential revisions:- The conclusion that the acto-myosin network contraction positions chromosomes for attachment to the spindle, as stated in the title, is not well substantiated. The authors do not directly assess chromosome attachment. The initial reduction of CSV (during the first ~8 min) remains intact in LINC-DN or pnBB-treated cells, and this is the time when chromosomes are expected to attach to the spindle. Unless the authors provide direct evidence, this conclusion needs to be toned down.

We found that, upon LINC-DN expression or pnBB treatment, chromosome congression to the middle of the mitotic spindle was significantly delayed (Figure 5B and our newly added Figure 5—figure supplement 3B). To achieve efficient chromosome congression, chromosomes must promptly establish initial interaction with spindle MTs and subsequently move from a position near the spindle poles to a position at the middle of the spindle. A delay in either step would lead to inefficient congression. In this context, we agree with the reviewers, and we presume that LINC-DN expression and pnBB treatment mainly delayed chromosome movement to middle of the spindle, rather than the initial chromosome interaction with spindle MTs. Consistent with this notion, LINC-DN expression and pnBB treatment diminished CSV reduction during 8-16 min after NEBD (Figure 2D and 3A), however at this stage, the initial chromosome–MT interaction had likely already occurred (as pointed out by reviewers) – for example, at 6–8 min after NEBD, several chromosomes had moved inward away from the NE remnants, probably due to MT interaction (Figure 2—figure supplement 2). Moreover, the appearance of back-of-spindle chromosomes after LINC-DN expression was well correlated with a delay in chromosome congression (Figure 5—figure supplement 2). We reason that, during 8–16 min after NEBD, the contractile acto-myosin network constrained chromosomes within the pole-to-pole region (Figure 3—figure supplement 6), thus reducing the occurrence of back-of-spindle chromosomes (Figure 5D and Figure 5—figure supplement 3D) and facilitating chromosome movement to the middle of the spindle. We have included this argument in Discussion of our revised manuscript.

Nonetheless the CSV reduction also occurs during an earlier phase (0–8 min after NEBD), and it is very possible that the earlier CSV reduction facilitates the initial chromosome–MT interaction. We would very much like to address this possibility. However, we currently lack the ability to inhibit the CSV reduction during this phase and to analyse consequent chromosome–MT interactions. In the current work, although we analysed timing of chromosome congression (Figure 5B and Figure 5—figure supplement 3B), we did not directly analyse earlier events of chromosome attachment to spindle MTs, leading to congression. Thus, as suggested by reviewers, we have toned down the relevant language and conclusion – for example, we used words ‘Evidence that…’ for a section title and ‘likely’ for conclusion in Results in our revised manuscript.

- An alternative explanation for the observed attachment defects is that microtubule-related mechanisms of congression are disrupted. Spindles in pnBB-treated cells clearly have an atypical morphology (Figure 3—figure supplement 3 and Figure 3—figure supplement 6) and it remains unclear whether this is secondary to attachment problems or could be the primary cause of the attachment defects. The authors should examine mitotic spindles in the LINC-DN and in jasplakinolide-treated cells to determine whether the spindle defects co-occur with the failure to reduce CSV. Additional visualization of a kinetochore marker would allow the authors to judge when chromosome attachment to the spindle happens in these cells.

As suggested, we have now analysed spindle morphology during the early stage of bipolar-spindle formation, in LINC-DN-expressing cells and in control cells. As indicated by the reviewers’ comment, pnBB-treated cells sometimes showed atypical spindle morphology, which includes (a) spindle MTs extending from a pole in wide-ranging directions (Figure 3—figure supplement 3, pnBB, +8 and +10 min) and (b) a pronounced bundle of interpolar MTs (Figure 3—figure supplement 6, pnBB, +10 min). However, such atypical spindle morphology was also found in LINC-DN-expressing cells and in control cells during the early stage of bipolar-spindle formation (newly added Figure 3—figure supplement 5). Thus, we did not find any obvious difference in the spindle morphology at this early stage of mitosis between control cells, pnBB-treated cells and LINC-DN expressing cells.

We have also visualised kinetochores and MTs using fluorescent protein tags and live-cell imaging. While we could identify kinetochore interaction with the bundles of spindle MTs, we were not able to confidently identify initial kinetochore–MT interaction, which is known to frequently occur on the lateral side of a single MT. To bypass this technical difficulty, we resorted to estimating chromosome interactions with spindle MTs by analysing chromosome positions relative to the actin network on the NE remnants (Figure 2—figure supplement 2. Soon after NEBD, chromosomes were found positioned adjacent to the acto-myosin network on NE remnants (Figure 2—figure supplement 2A, during 1–4 min; Figure 2—figure supplement 2B). We reasoned that the contractile acto-myosin network was pushing chromosomes inward during this period. Subsequently (6–8 min following NEBD), in the region where the bipolar spindle was formed, those chromosomes directly adjacent to the acto-myosin network moved away from the network and thus were no longer spatially associated with the network (Figure 2—figure supplement 2C). We interpreted that chromosomes established interactions with spindle MTs, and subsequently MT-based mechanisms had moved the chromosomes further inward, away from the acto-myosin network. Though this method does not provide precise timing of initial chromosome–MT interaction, we can estimate that the interaction largely occurred within 8 min following NEBD.

- The text is very concise, which makes it easy to read. However, the reviewers felt that it was too succinct, and that the discussion of prior work was not given adequate space. Prior work on the position and movement of chromosomes in early mitosis, on the relation between actin/myosin and spindles and on the 'spindle envelope' seems relevant. Papers that were mentioned as important by the reviewers include: Baarlink et al., NCB 2017; Barisic et al., 2014; Itoh et al., 2018; Khodjakov work on chromosome congression; Kunda and Baum, 2009; Whitehead et al., 1996; Rosenblatt et al., 2004; Heng et al., J Cell Sci 2012; Woolner et al., 2008; papers by A. Forer (Protoplasma 2007, Cytoskeleton 2013,.…).

As suggested, we now have extended the Introduction and Discussion sections in our revised manuscript. We now cite and discuss most of these papers, suggested by reviewers, in the second paragraph of the revised Introduction and in the fourth paragraph of the revised Discussion.

- In many experiments, the authors only analyzed a small number of cells, and it seems that control cells from different days yield average normalized CSV curves that are significantly different from each other. For the most part, the differences between control- and treated cells seem large enough that this is not a concern. However, the conclusion that nocodazole-treated cells differ at the 4-min time point from control cells (Figure 1C) does not seem robust. Picking different control cells would yield a different conclusion. The authors either need to image more cells or tone down their conclusion. The number of asynchronously growing cells observed for CSV at mitotic entry (Figure 1—figure supplements 1-4) should be increased beyond three.

We have now toned down our conclusion regarding the 4-min time point in Figure 1C by stating that ‘CSV may be slightly smaller at 4 min following NEBD in control cells; if so, this could be due to…’ (words in bold were inserted in the revised manuscript).

We have also now increased the number of cells in our analysis, from three to eight, in Figure 1—figure supplement 2. Our conclusion from the extended data set remains unchanged.

- Since the authors speculate about potential segregation defects in the absence of the CSV-reduction mechanism, it would be appropriate to analyze chromosome segregation in the LINC-DN or pnBB cells, which is easily feasible.

As suggested, we have now scored frequency of abnormal chromosome segregation in control, LINC-DN-expressing and pnBB-treated cells (Figure 5E and Figure 5—figure supplement 3E). In 18–20% of LINC-DN expressing and pnBB-treated cells, abnormal chromosome segregation was observed, i.e. chromosomes were left either in the middle of spindle or at the back of spindle during anaphase.